# Plasma Leptin and Adiponectin after a 4-Week Vegan Diet: A Randomized-Controlled Pilot Trial in Healthy Participants

**DOI:** 10.3390/ijerph191811370

**Published:** 2022-09-09

**Authors:** Ann-Kathrin Lederer, Maximilian Andreas Storz, Roman Huber, Luciana Hannibal, Elena Neumann

**Affiliations:** 1Center for Complementary Medicine, Department of Medicine II, Medical Center-University of Freiburg, Faculty of Medicine, University of Freiburg, 79106 Freiburg, Germany; 2Department of General, Visceral, and Transplant Surgery, University Medical Center Mainz, 55131 Mainz, Germany; 3Laboratory of Clinical Biochemistry and Metabolism, Department of General Pediatrics, Adolescent Medicine and Neonatology, Medical Center-University of Freiburg, Faculty of Medicine, University of Freiburg, 79106 Freiburg, Germany; 4Department of Rheumatology and Clinical Immunology, Campus Kerckhoff, Justus-Liebig-University Giessen, 61213 Bad Nauheim, Germany

**Keywords:** hormones, diet, nutrition, branched-chain amino acids, plant-based diet, sex, obesity, leptin/adiponectin ratio

## Abstract

Adiponectin and leptin are important mediators of metabolic homeostasis. The actions of these adipokines extend beyond adipocytes and include systemic modulation of lipid and glucose metabolism, nutrient flux, and the immune response to changes in nutrition. Herein, we hypothesized that short-term intervention with a vegan diet might result in an improvement of plasma concentrations of adiponectin and leptin and the leptin/adiponectin ratio. We investigated the response of plasma adiponectin and leptin to a 4-week intervention with a vegan or meat-rich diet and its associations with sex, BMI and nutritional intake. Fifty-three healthy, omnivore participants (62% female, average age 31 years and BMI 23.1 kg/m^2^) were randomly assigned to a vegan or meat-rich diet for 4 weeks. Plasma adiponectin and leptin were lower in men compared to women both at the beginning and end of the trial. The concentration of adiponectin in women was significantly higher both when comparing their transition from omnivorous to vegan diet (*p* = 0.023) and also for vegan versus meat-rich diet at the end of the trial (*p* = 0.001), whereas plasma leptin did not vary significantly. No changes in adiponectin were identified in men, yet an increase in leptin occurred upon their transition from an omnivorous to a meat-rich diet (*p* = 0.019). Examination of plasma adiponectin/leptin ratio, a proposed marker of cardiovascular risk, did not differ after 4-weeks of dietary intervention. Our study revealed that adiponectin and leptin concentrations are sensitive to short-term dietary intervention in a sex-dependent manner. This dietary modification of leptin and adiponectin not only occurs quickly as demonstrated in our study, but it remains such as published in studies with individuals who are established (long-term) vegetarians compared to omnivorous.

## 1. Introduction

Up to a third of the body mass of a healthy, normal-weight human being consists of white adipose tissue, which is able to influence major body functions including metabolism, immune system and regulation of circulation [1]. The white adipose tissue produces cytokines, so-called adipokines, which contribute to regulating appetite and energy homeostasis, but are also known to affect inflammatory processes and the functionality of the immune system [2,3,4]. Several adipokines are known, but leptin and adiponectin are the most popular ones being the focus of thousands of scientific publications. The circulating adipokine concentration depends on the amount of white adipose tissue [5]. The concentration of leptin is positively correlated with the BMI [6,7], and the concentration of adiponectin is negatively correlated with the BMI [8]. The concentration of adipokines also depends on sex as women have greater leptin concentrations than comparable men [9,10]. Normally, low levels of leptin are responsible for feeling hungry, whereas high levels decrease appetite [11]. Leptin has essential immune regulatory functions being involved in the proliferation and differentiation of different types of immune cells [3,5,12,13]. Congenital leptin deficiency can lead to a life-threatening risk of infection, but also to disturbances of the glucose metabolism, severe dyslipidemia, endocrine disruptions and developmental disabilities [14,15]. Leptin was also found to be increased in patients with auto-inflammatory diseases such as rheumatoid arthritis [5,16]. Adiponectin is reported to be associated with disease activity of patients with rheumatoid arthritis [16,17], and low adiponectin concentrations are associated with metabolic syndrome, obesity-linked insulin resistance and atherosclerosis [8,18,19]. The ratio of adiponectin and leptin is reported to be a promising index to estimate adipose tissue inflammation and cardiovascular risk [20,21].

Adipose tissue-driven chronic low-grade inflammation has been proposed as an underlying mechanism for the development of a vast array of chronic diseases [1]. Plant-based diets are reported to reduce markers of chronic inflammation, improve symptoms of anti-inflammatory diseases and lower cardiovascular morbidity and mortality [22,23,24,25,26,27,28,29]. Long-term vegan and vegetarian diets might affect leptin and adiponectin concentrations of normal-weight individuals [29,30,31]. Despite the established interplay of metabolism and immune function, the underlying mechanism of these diet-induced anti-inflammatory effects in humans has been insufficiently studied. In 2017, we started a clinical trial aiming to broadly map the influence of a short-term vegan diet on the immune system to elucidate the underlying mechanism leading to an anti-inflammatory effect of a plant-based diet [32,33]. Herein, we hypothesized that short-term intervention with a vegan diet in healthy, normal-weight omnivorous subjects might result in an improvement of the plasma concentrations of the metabolic mediators, adiponectin and leptin, and the ratio of adiponectin/leptin. With this exploratory analysis of adipokines, we aimed to estimate the changing of adipokines after starting a vegan diet compared to a meat-rich diet.

## 2. Materials and Methods

A monocentric, controlled, randomized trial with a parallel group design with healthy participants was performed at the Center for Complementary Medicine, University Medical of Freiburg, Germany, between April and June 2017. The analysis of leptin and adiponectin is a subgroup analysis of a clinical trial evaluating the effect of a vegan diet on the immune system, vitamin B-12 metabolism and gut microbiome, which is why methods were previously published in detail [32,33]. Before onset, the trial was approved by the ethics committee of the University Medical Center of Freiburg, Germany (EK Freiburg 38/17), and was registered at the German Clinical Trial Register (DRKS00011963). The trial was performed according to the principles of the declaration of Helsinki.

Criteria of inclusion were: Healthy, normal-weight or overweight subjects (BMI between 18.5 and 30 kg/m^2^), being on a free-of-choice omnivorous diet, between 18 and 60 years of age, no regular intake of medication as well as no clinically relevant allergies. Participants declared having no history of an eating disorder, participation in another clinical trial and blood donation in the last 4 weeks before the start of this trial. Abuse of drugs, nicotine or alcohol was not allowed. Participants had to be able to speak and understand German and to complete a nutritional protocol.

Newspaper announcements and bulletins were used for recruitment. After a phone call, eligible subjects were invited for a personal visit to check eligibility criteria in detail. A precondition for inclusion was written informed consent. After inclusion, each participant received extensive training to keep his/her own balanced mixed omnivorous diet according to the recommendations of the German Nutrition Association (DGE) [34]. After this one-week-lasting run-in phase, fasting (>6 h) baseline parameters were taken early in the morning. Afterwards, participants were randomly assigned to either a meat-rich (>150 g of meat per day; any meat of their choice) or a strict vegan diet for four weeks. Every subject received extensive training on the assigned diet and detailed written information and a recipe book. Fasting parameters were taken again after four weeks early in the morning. Participants finished the trial after blood sampling. No pre-cooked meals were served. Participants were free to choose their food within their assigned diet regardless of whether the food was healthy or not, but filling out a weekly nutritional protocol was mandatory for all participants. Results of nutritional protocols were used to evaluate diet adherence. Additionally, all participants had to control their weight daily as weight changes of more than 2 kg were not allowed. Weekly follow-ups were scheduled between the study staff and the participants by phone or e-mail. Participants, who reported about weight loss, were recommended to eat more high-caloric foods (e.g., nuts), whereas participants with weight gain were recommended to avoid high-calorie foods.

Serum from each participant was collected and analyzed blinded for diet assignment. Aliquots were prepared in 1.5-mL cryovials without information about the type of diet on the labels and stored at −20 °C until dry-iced cooled transport to the Department of Rheumatology and Clinical Immunology, Bad Nauheim, Germany. Adiponectin and leptin proteins were measured by ELISA as described by the manufacturer (Quantikine ELISA Human DRP300, DLP00 Bio-Techne R&D Systems, Wiesbaden, Germany).
$$\begin{array}{l} \mathrm{Assay}\;\mathrm{Range}:\\ \mathrm{Adiponectin}\;3.9–250\;µg/\mathrm{mL},\;\mathrm{sensitivity}\;0.891\;\mathrm{ng}/\mathrm{mL}.\\ \mathrm{Leptin}\;15.6–1000\;\mathrm{pg}/\mathrm{mL},\;\mathrm{sensitivity}\;7.8\;\mathrm{pg}/\mathrm{mL}. \end{array}$$


Measurement of amino acids was performed by the Laboratory of Clinical Biochemistry and Metabolism, Department for Pediatrics, University Hospital of Freiburg [35]. Measurement of leukocytes, monocytes and C-reactive protein was performed by the Central Laboratory of the University Hospital of Freiburg.

The randomization list was created electronically block-wise (block size 13; Python Software, Python Software Foundation, Wilmington, DE, USA) by a third independent person, sealed envelopes were used for implementation, and randomization was assigned directly after taking baseline blood samples by two members of the study group.

### Statistical Analysis

The analysis of leptin and adiponectin was an exploratory aim. As mentioned before, this is a subgroup analysis of a clinical trial evaluating the effect of a vegan diet on the immune system, vitamin B-12 metabolism and gut microbiome [32,33]. Therefore, the sample size was planned for the main parameters of the pilot trial. Considering a statistical power of 80% and a hypothesized large effect size of 1 it was calculated that 48 participants (24 on a vegan diet and 24 on a meat-rich diet) would be needed to detect a statistical difference of *p* < 0.05 between the groups. To evaluate the effect of Time, Diet and Sex and their interactions on the levels of leptin and adiponectin, with estimated large effect size, at a significant alpha level of 0.05, and the number of participants was equal to 53 (men: 8 vegan diet and 12 meat-rich diets; women: 18 vegan diet and 15 meat-rich diet) a post hoc statistical power of 81% was calculated. Power analysis was calculated using G*Power (version 3.1.9.7, University of Düsseldorf, Düsseldorf, Germany).

Data were entered and analyzed blinded for diet assignation via IBM SPSS (version 27.0, IBM, Armonk, NY, USA). Baseline characteristics were evaluated by *t*-test, Mann-Whitney-U-Test and Fisher’s exact test. Because some biochemical markers were not normally distributed, Mann-Whitney-U-test was used for comparison of group differences. Adjustment for baseline was considered by using ANCOVA. Multiple linear regression was performed to evaluate the dependency effects of age, BMI (end concentration), weight changing (weight change of more than one SD (±1.3 kg) compared to baseline weight), sex and diet on the end concentration of leptin and adiponectin. A three-way mixed ANOVA was used for evaluating the between-subject effects of diet (vegan diet, meat-rich diet) and sex (male, female), the within-subject effect of time (baseline, end), and their interactions.

## 3. Results

Out of 150 interested persons, 53 participants were eligible for inclusion and were willing to participate; 26 participants were randomly allocated to a vegan diet and 27 participants were randomly allocated to a meat-rich diet for four weeks. The flow of participants is shown in Figure 1. All participants completed the study as per protocol. At both time points (baseline and end), BMI did not differ significantly between the groups. The intake of energy was similar in both groups (Table 1) and was within the recommendations set forth by the DGE for healthy adults [34]. Demographic data of included participants are shown in Table 1.

### 3.1. Systemic Leptin and Adiponectin before and after Vegan and Meat-Rich Diet

Overall comparison of baseline and end concentrations of leptin as well as of adiponectin between the vegan diet group and meat-rich diet group did not reach statistical significance (Table 2). Adiponectin/leptin ratio did not change significantly during the trial (*p* = 0.55). The adiponectin/leptin ratio did not differ significantly between vegan diet and meat-rich diet (baseline vegan diet vs. meat-rich diet, *p* = 0.55; end vegan diet vs. meat-rich diet, *p* = 0.18). To verify the validity of overall data, we performed analysis of sex as a well-known influencing factor on adipokine concentration. As expected, results of leptin and adiponectin measurement were significantly affected by participants’ sex as men had significantly lower levels of leptin (baseline men 5.7 ± 4.2 vs. women 23.8 ± 11.7 ng/mL, *p* = 0.002; end men 6.9 ± 4 vs. women 22.1 ± 13.6 ng/mL, *p* = 0.001) and adiponectin (baseline men 9.1 ± 5.2 vs. women 23.8 ± 11.7 ng/mL, *p* < 0.001; end men 9.3 ± 6.2 vs. women 22.1 ± 13.6 ng/mL, *p* < 0.001) in compared to women. The adiponectin/leptin ratio differed significantly between women and men at the beginning of the trial (baseline men 1.9 ± 1.9 vs. women 0.82 ± 0.7, *p* = 0.003; end men 1.7 ± 1.5 vs. women 1 ± 0.8, *p* = 0.1). The effect of multiple predictors on end concentration of leptin and adiponectin is shown in Table 3. The end concentration of leptin depended on participants’ end BMI as well as on diet and sex. The end concentration of adiponectin depended on the participants’ end BMI.

### 3.2. Influence of Diet on Leptin and Adiponectin Levels of Women and Men

As the concentration of leptin and adiponectin was significantly affected by participants’ sex (see above), women and men were analyzed separately.

#### 3.2.1. Influence of Diet on Leptin and Adiponectin Levels of Women

Women in the vegan diet group (*n* = 18) lost on average 0.6 kg (range −2.2–3.4 kg; baseline 64.4 kg, end 63.9 kg, *p* = 0.033) body weight during the trial. In contrast, women in the meat-rich diet group (*n* = 15) had a non-significant weight gain of 0.4 (range −0.8–2 kg; baseline 61.7 kg, end 62.1 kg, *p* = 0.17). BMI of the vegan diet group was 22.6 ± 1.9 kg/m^2^ before the trial and 22.4 ± 1.8 kg/m^2^ at the end of the trial (*p* = 0.033). BMI of the meat-rich diet group was 22.6 ± 2.8 kg/m^2^ before the trial and 22.8 ± 2.7 kg/m^2^ at the end of the trial (*p* = 0.16).

Comparison of baseline and end concentration of leptin between a vegan diet and meat-rich diet female subgroups did not reach statistical significance (Table 4 and Figure 2, Panel A). The end concentration of leptin depended on female participants’ end BMI (*p* < 0.001, Table 4).

A comparison of baseline and end concentration of adiponectin showed a significantly higher concentration in the female vegan diet subgroup at the end of the trial (*p* = 0.023, Table 4 and Figure 2, Panel B). Furthermore, the end concentration of adiponectin was significantly higher in a vegan diet than in a meat-rich diet (18.8 vs. 12.7 µg/mL, *p* = 0.001). The end concentration of adiponectin depended on diet in women (*p* = 0.01, Table 5). The results of mixed ANOVA revealed an interaction effect between time and diet (time × diet *p* = 0.004) regarding the adiponectin concentration (Table 4).

#### 3.2.2. Influence of Diet on Leptin and Adiponectin Levels of Men

Men in the vegan diet group (*n* = 8) had a non-significant weight loss of 0.5 kg (range −2.2–2.2 kg; baseline 77.1 kg, end 76.6 kg, *p* = 0.18) during the trial. Men in the meat-rich diet group (*n* = 12) had a non-significant weight gain of 0.4 kg (range −2.9–2.3 kg; baseline 80.3 kg, end 80.7 kg, *p* = 0.21). BMI of the vegan diet group was 23.6 ± 2.7 kg/m^2^ before the trial and 23.4 ± 2.5 kg/m^2^ at the end of the trial (*p* = 0.208). BMI of meat-rich diet group was 24.1 ± 2.2 kg/m^2^ before the trial and 24.2 ± 2.3 kg/m^2^ at the end of the trial (*p* = 0.21).

Leptin concentration increased significantly during the trial in the meat-rich diet group (*p* = 0.019, Table 6 and Figure 3, Panel A). The results of mixed ANOVA did not show any main effects of time or diet as well as interaction effects between diet and time on leptin concentration in men (Table 6). In men, multiple linear regression did not reveal any dependency of BMI on the end concentration of leptin (Table 7).

Adiponectin concentration remained stable during the trial in men (Table 8 and Figure 3, Panel B). The results of mixed ANOVA did not show any main effects of time or diet as well as interaction effects between diet and time on adiponectin concentration in men (Table 6). Multiple linear regression did not reveal any dependency on end concentration of adiponectin (Table 7), but the results are limited due to the poor test quality.

### 3.3. Impact of Nutritional Intake on Leptin and Adiponectin Concentration

Nutritional intake of fat, protein, cholesterol and fatty acids differed significantly between a vegan diet and a meat-rich diet (Table 8). We found no significant sex-specific differences regarding participants’ nutritional intake.

## 4. Discussion

This pilot study was driven by the hypothesis that the adipokines leptin and adiponectin contribute to the immunomodulatory effects observed in an earlier study upon the intervention of healthy subjects with a short-term balanced vegan diet. The results of our study did not show a clear diet-related changing of leptin and adiponectin after a 4-week lasting vegan diet vs. a meat-rich diet. Interestingly, we found a sex-dependent response of these adipokines to dietary changing. The comparison of baseline and end concentration in meat-rich diet participants revealed a significant increase of leptin in men, whereas adiponectin changed within the vegan diet group only in women. The results are limited due to the exploratory character of the study and the small sample-size, but thorough trial implantation considering recent scientific standards strengthens the trial and indicates ideas for further research. Due to the weekly contact with our participants, the control of nutritional protocils and based on the previously published results of the vitamin B-12 metabolism, we are convinced of the diet adherence of our subjects [32]. Basically, we found that the participants’ BMI had an impact on leptin concentration emphasizing good data quality as the relationship between leptin and BMI is well-known [7]. Good data quality is also supported by the well-known sex-dependent differences of adipokines as well as the adiponectin/leptin ratio in men and women, which is also observed in our trial [10,36,37,38,39,40]. Interestingly, the results of our and other studies emphasize not only the previously known fundamental difference in adipokine levels in men and women, but also a supposed sex-difference in alterability by diet or specific nutritional components [41]. This is also supported by the results of Vučić Lovrenčić et al. as they observed differences of adiponectin concentration in female vegetarians compared to female omnivores, but not in male vegetarians compared to male omnivores [39]. Leptin is an immunomodulatory cytokine, which was observed to be increased in patients with rheumatic diseases [16,42]. Consumption of meat is hypothesized to be one of the major contributors to promoting inflammation in rheumatic diseases being supported by clinically observed improvement of disease activity by plant-based diets [23,24,43]. The increase of meat intake in our male meat-rich diet participants led to an increase of leptin which supports a potential relationship between higher meat intake and inflammation. It remains unclear whether an increase in leptin leads to pathologically increased inflammation in healthy participants, as none of the participants showed adverse effects clinically or in laboratory tests [33]. Astonishingly, female meat-rich diet participants did not show an alteration of leptin in turn emphasizing the sex-dependent response to diet changing. The evaluation of leptin and adiponectin changing in women is challenging as women are also affected by cyclic sexual hormone changing [9]. The menstrual cycle was not captured in our trial, which implies a potential bias of our results concerning especially women [44]. Furthermore, despite the presetting of avoid any weight changing, women in the vegan diet group had a slight weight-loss, which might confound the increase in adiponectin after the trial since the difference might not be caused by diet, but by weight loss. Weight-loss is discussed to be related to changing in adipokine levels. Published reports indicate the effects of weight loss on serum leptin, but the same has not been unequivocally seen with respect to serum adiponectin concentration [45]. Therefore, we performed a subgroup analysis excluding participants with a weight change of more than 1.3 kg (more than one SD). Interestingly, the difference of adiponectin between a vegan diet and a meat-rich diet remained statistically significant, suggesting a not-weight related effect of diet on adiponectin levels in women. In men, who had no weight-loss during the trial, adiponectin remained stable, but the comparability of men and women is limited due to the previously mentioned aspects.

Apart from sex-related differences, the general impact of specific diets or of specific nutritional components on adipokines levels of normal-weight human beings remains still unclear, since the majority of publications dealt with adipokine changes in weight-losing subjects or in animals [46]. The comparison of a controlled 12-week-lasting protein-rich vegan diet revealed higher levels of adiponectin in vegan rats compared to omnivorous rats [47]. However, the choice of food, whether healthy or not, might be crucial for adiposity-associated biomarker concentrations such as leptin and adiponectin [48,49]. In our trial, the total intake of calories was similar in the vegan diet and meat-rich diet, but the components of the diet especially of proteins and fats differed significantly. It is, therefore, likely that the changing of diet composition might be responsible for the observed effects. In vitro studies indicate for example a decrease of leptin by virgin olive oil components making it a potentially promising therapeutic agent for the treatment and prevention of obesity [50]. A systematic review by Eichelmann et al. indicated no pronounced effect of plant-based diets on adipokines levels, but they added that the analysis was restricted by the number and quality of available studies suggesting necessity for more research [22]. Some other clinical trials indicated that specific nutritional components such as soups, vegetables, vegetable oils or dietary fiber might have an increasing effect on leptin concentration in weight-stable healthy participants [41,51,52,53]. Havel et al. hypothesized that greater intake of fat might lower leptin concentration whereas a low fat/high carb diet induces higher levels of leptin. Rodent models have shown an effect of dietary fatty acids on serum adiponectin indicating lower levels of adiponectin in high fat diet supporting our results, but clinical data did not confirm the results [54,55].

Initially, we hypothesized an alteration of the adiponectin/leptin ratio by diet, but this was not confirmed by our results. Contrary, recent research suggests a higher adiponectin/leptin ratio in planted-based diets implying a decreased cardiovascular risk. Ambroszkiewicz et al. reported about vegetarian children, who showed a higher adiponectin/leptin ratio compared to omnivorous children [56]. In a follow-up trial of the same research group, they did not find a significant difference of adiponectin levels in vegetarian and omnivorous children [57]. Recent literature suggests pro- and anti-inflammatory effects of adiponectin depending on pre-existing diseases [58]. In participants not suffering from inflammatory diseases, adiponectin appears to have the anti-inflammatory potential being an insulin-sensitizing, vascular-protective and anti-inflammatory protein [16,58,59]. Similar effects are described for the vegan diet underlining a potential relation between higher levels of adiponectin in the vegan diet [60,61]. A cross-sectional trial by Menzel et al. did not reveal differences in adiponectin levels between vegans and omnivores [31]. Furthermore, a systematic review of the same research group were not able to demonstrate a clear difference of adiponectin between a plant-based diet and an omnivorous diet [29]. The vegan diet is reported to lower inflammation and cardiovascular risk, but this does not seem to be reflected by adipokine alteration after the diet change. Recent research data are really inhomogeneous making it hard to draw a firm conclusion about adipokine changes in plant-based diets. Furthermore, the definition of a plant-based diet is often inconsistent across publications, which must be considered when evaluating the results of nutritional trials [62]. Diets in cross-sectional studies are not generally health-driven diets. The diet in our trial was also not health-driven as, participants were able to choose their food within their assigned diet, but without instruction to avoid unhealthy food. As mentioned before, components of food appear to be crucial for adipokine changing and not the fact of eating plant-based.

The relatively short duration of our trial, 4 weeks, is a blessing and a curse as it pictures early alteration of leptin and adiponectin after diet change, but also contributes to the apparent diffuse association between diet and adipokine profiles. Interestingly, leptin is reported to respond quickly to changes in dietary composition. As mentioned before, more than 20 years ago, Havel et al. performed a trial comparing a just 24-h-lasting high fat/low carb vs. a low fat/high carb diet in healthy women, and found a decrease of the physiologically circadian rhythmic leptin concentration in participants with high fat/low carb diet [63]. This finding of leptin is supported by other studies indicating possibility of rapid modulation of human adipokine levels [9,64,65].

Overall, as a clinical consequence of our trial, the results indicate that nutrition appears to be not a “one size fits all” concept also emphasizing the individual as well as the sex-dependent response to diet.

## 5. Conclusions

In summary, serum levels of the most commonly known adipokines, leptin and adiponectin, do not fully explain the immunomodulatory potential of the vegan diet in healthy participants. However, the results of our trial suggest that the effect of the vegan diet and meat-rich diet on leptin and adiponectin levels might depend on participants’ sex, as men and women showed different responses on the nutritional change. Elucidating whether the observed sex-specific differences emerge from the inflammatory potential of diets requires further investigation, ideally in long-term trials.

## Figures and Tables

**Figure 1 ijerph-19-11370-f001:**
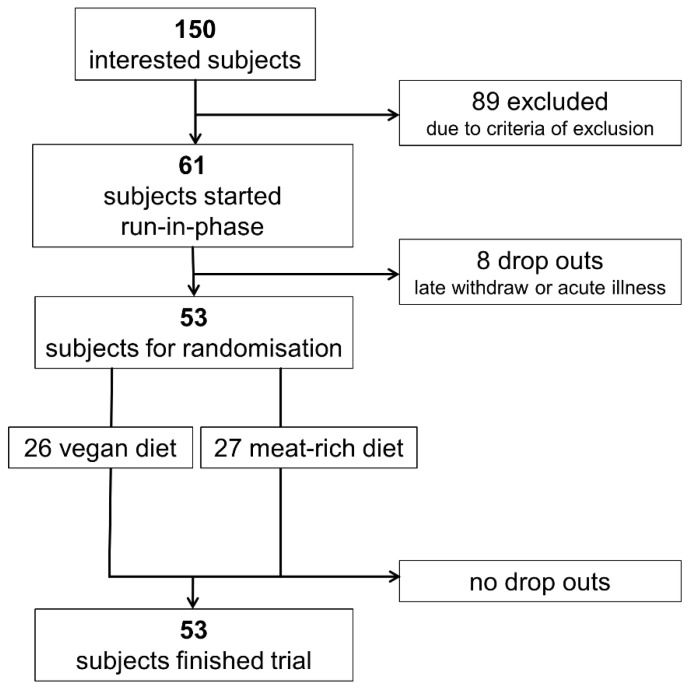
Flowchart visualizing recruitment of participants.

**Figure 2 ijerph-19-11370-f002:**
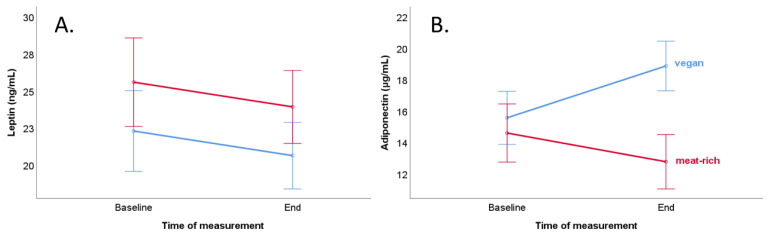
Course of leptin and adiponectin levels of the vegan group (blue line) and the meat-rich group (red line) during the trial with consideration of potential confounders (BMI and age) in women. Bars show standard error ± 1. (**A**) Course of serum leptin concentration (ng/mL). (**B**) Course of serum adiponectin concentration (µg/mL).

**Figure 3 ijerph-19-11370-f003:**
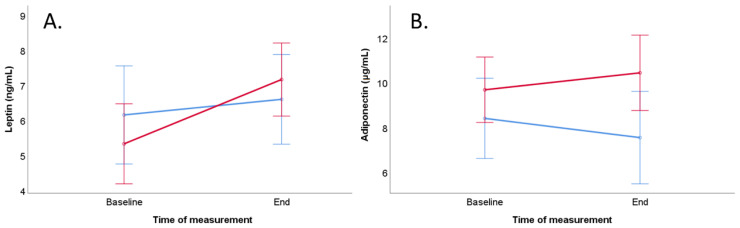
Course of leptin and adiponectin levels of the vegan group (blue line) and meat-rich group (red line) during the trial with consideration of potential confounders (BMI and age) in men. Bars show standard error ± 1. (**A**) Course of serum leptin concentration (ng/mL). (**B**) Course of serum adiponectin concentration (µg/mL).

**Table 1 ijerph-19-11370-t001:** Demographic data of all participants in the vegan diet group and in the meat-rich diet group.

	Vegan (*n* = 26)	Meat-Rich (*n* = 27)	*p* * (VeD vs. MrD)
Age (years)	33.2 ± 11.2	29.9 ± 9.5	0.41
Sex (n male/n female)	8/18	12/15	0.23
Intake of energy (kcal)	2240 ± 894	2242 ± 762	0.21
*Body mass index (kg/m*^2^*)*			
Baseline	22.9 ± 2.2	23.3 ± 2.6	0.44
End of study	22.7 ± 2	23.4 ± 2.6	0.24
*Weight (kg)*			
Baseline	68.3 ± 10.5	70 ± 13.3	0.76
End of study	67.8 ± 10.2	70.3 ± 13.2	0.59

Results are presented as average value ± standard deviation; VeD = vegan diet, MrD = meat-rich diet. * *p*-value from *t*-test for parametric continous values (BMI baseline)/Mann-Whitney-U-Test for non-parametric continous values (age, intake of engery, BMI end, weight baseline and end)/Fisher’s exact test for categorial variables (sex).

**Table 2 ijerph-19-11370-t002:** The serum concentration of leptin and adiponectin before and at the end of the trial in and between both groups as well as main and interaction effects.

		Leptin (ng/mL)		Adiponectin (µg/mL)	
		Baseline	End	Baseline	End
Vegan (VeD) ^†^		17.3 ± 11.2	15.8 ± 11.5	13.4 ± 6.9	15.5 ± 7.6
Meat-rich (MrD) ^†^		16.6 ± 14.7	16.9 ± 14.8	12.3 ± 7.2	11.6 ± 7.3
**VeD** (Baseline vs. end value) *p* *		0.16		0.06	
**MrD** (Baseline vs. end value) *p* *		0.39		0.4	
**VeD****vs. MrD** (Baselines) *p* **^+^**		0.55		0.34	
**VeD****vs. MrD** (End values) *p* **^+^**			0.6 ^§^		**0.025** ^§^
**Main & interaction effects °**					
*p*-value	time	0.88		0.6	
	diet	0.43		0.58	
	sex	**<0.001**		**0.001**	
	time × diet	0.64		0.17	
	time × sex	0.37		0.56	
	time × diet × sex	0.99		**0.014**	

Results are presented as average value ± standard deviation; MrD = Meat-rich diet, VeD = Vegan diet. ^†^ vegan diet group *n* = 26, meat-rich diet group *n* = 27, * Wilcoxon signed-rank-test, ^+^ Mann-Whitney-U-Test, ° Mixed ANOVA, ^§^ adjusted for baseline. Significant *p*-values are shown in bold.

**Table 3 ijerph-19-11370-t003:** Associations between leptin and adiponectin with age, BMI, diet and changing of weight.

	Leptin				Adiponectin			
	β	95% CI		*p* ^+^	β	95% CI		*p* ^+^
		Upper	Lower			Upper	Lower	
Age	−0.118	−0.4	0.1	0.2	0.106	−0.1	0.3	0.4
BMI	0.702	13.9	24	**<0.001**	0.347	1.4	9.4	**0.01**
Diet	0.57	2.1	4.3	**<0.001**	−0.238	−1.6	0.1	0.07
Changing of weight *	0.034	−4	5.7	0.71	−0.17	−6.4	1.3	0.19

^+^ Multiple linear regression of end concentration of leptin (R2 = 0.629, Cohen’s f2 = 1.69, post hoc power analysis = 100%) and of adiponectin (R2 = 0.298, Cohen’s f2 = 0.43, post hoc power analysis = 95%), significant *p*-values are shown in bold; * “Changing of weight” is defined as weight change of more than one standard deviation (±1.3 kg) compared to baseline weight; β = standardized beta coefficient.

**Table 4 ijerph-19-11370-t004:** The serum concentration of leptin and adiponectin before and at the end of the trial in and between both groups, as well as main and interaction effects in women.

		Leptin (ng/mL)		Adiponectin (µg/mL)	
		Baseline	End	Baseline	End
Vegan ^†^ (VeD)		22.3 ± 9	20.2 ± 11.1	15.5 ± 6.2	18.8 ± 6.5
Meat-rich ^†^ (MrD)		25.4 ± 14.4	24.6 ± 16.1	14.6 ± 8.2	12.7 ± 7
**VeD** (Baseline vs. end value) *p* *		0.16		**0.023**	
**MrD** (Baseline vs. end value) *p* *		0.96		0.17	
**VeD vs. MrD** (Baselines) *p* ^+^		0.68		0.42	
**VeD vs. MrD** (End values) *p* ^+^			0.51 ^§^		**0.001 ^§^**
**Main & interaction effects °**					
*p*-value	time	0.49		0.38	
	diet	0.33		0.13	
	time × diet	0.75		**0.004**	

Results are presented as average value ± standard deviation; MrD = Meat-rich diet, VeD = Vegan diet. ^†^ vegan diet group *n* = 18, meat-rich diet group *n* = 15, * Wilcoxon signed-rank-test, ^+^ Mann-Whitney-U-Test, ° Mixed ANOVA, ^§^ adjusted for baseline. Significant *p*-values are shown in bold.

**Table 5 ijerph-19-11370-t005:** Associations between leptin and adiponectin with age, BMI, diet and changing of weight in women.

	Leptin				Adiponectin			
	β	95% CI		*p* ^+^	β	95% CI		*p* ^+^
		Upper	Lower			Upper	Lower	
Age	−0.251	−0.7	0.1	0.05	0.017	−0.2	0.3	0.92
BMI	0.802	3.2	6.4	**<0.001**	−0.289	−2.1	0.2	0.11
Diet	0.484	−5.1	8.2	0.63	−0.454	−11.4	−1.7	**0.01**
Changing of weight *	−0.055	−7.7	7.3	0.96	0.281	−1.1	9.9	0.12

^+^ Multiple linear regression of end concentration of leptin (R2 = 0.62, Cohen’s f2 = 1.63, post hoc power analysis = 100%) and of adiponectin (R2 = 0.29, Cohen’s f2 = 0.41, post hoc power analysis = 78%), significant *p*-values are shown in bold; * “Changing of weight” is defined as weight change of more than one standard deviation (±1.3 kg) compared to baseline weight, β = standardized beta coefficient.

**Table 6 ijerph-19-11370-t006:** Serum concentration of leptin and adiponectin before and at the end of the trial in and between both groups as well as main and interaction effects in men.

		Leptin (ng/mL)		Adiponectin (µg/mL)	
		Baseline	End	Baseline	End
Vegan ^†^ (VeD)		5.9 ± 6	6.3 ± 5	8.6 ± 6.1	7.9 ± 2.7
Meat-rich ^†^ (MrD)		5.5 ± 2.5	7.3 ± 3.4	9.5 ± 4.7	10.2 ± 7.7
**VeD** (Baseline vs. end value) *p* *		0.74		0.89	
**MrD** (Baseline vs. end value) *p* *		**0.019**		0.75	
**VeD****vs. MrD** (Baselines) *p* **^+^**		0.73		0.57	
**VeD****vs. MrD** (End values) *p* **^+^**			0.25 ^§^		0.43 ^§^
**Main & interaction effects °**					
*p*-value	time	0.06		0.97	
	diet	0.86		0.53	
	time × diet	0.24		0.45	

Results are presented as average value ± standard deviation; MrD = Meat-rich diet, VeD = Vegan diet. ^†^ vegan diet group *n* = 8, meat-rich diet group *n* = 12, * Wilcoxon signed-rank-test, ^+^ Mann-Whitney-U-Test, ° Mixed ANOVA, ^§^ adjusted for baseline. Significant *p*-values are shown in bold.

**Table 7 ijerph-19-11370-t007:** Associations between leptin and adiponectin with age, BMI, diet and changing of weight in men.

	Leptin				Adiponectin			
	β	95% CI		*p ^+^*	β	95% CI		*p ^+^*
		Upper	Lower			Upper	Lower	
Age	0.128	−0.1	0.2	0.51	0.370	−0.1	0.5	0.12
BMI	0.421	−0.1	1.5	0.06	−0.388	−2.4	0.4	0.14
Diet	0.045	−2.9	3.6	0.82	0.261	−2.7	9.1	0.26
Changing of weight *	−0.405	−6.8	0.3	0.07	−0.125	−8.1	5	0.62

^+^ Multiple linear regression of end concentration of leptin (R2 = 0.486, Cohen’s f2 = 0.946, post hoc power analysis = 86%) and of adiponectin (R2 = 0.273, Cohen’s f2 = 0.38, post hoc power analysis = 44%) in men; * “Changing of weight” is defined as weight change of more than one standard deviation (±1.3 kg) compared to baseline weight, β = standardized beta coefficient.

**Table 8 ijerph-19-11370-t008:** Results of nutritional protocols.

	Vegan (*n* = 22 ^+^)	Meat-Rich (*n* = 22 ^+^)	*p* *
Total intake of energy (kcal)	2240 ± 894	2242 ± 762	0.21
Kcal/kg body weight	32 ± 9.5	35 ± 12.3	0.37
Proteins/kg body weight	1 ± 4	1.6 ± 0.5	**<0.001**
Carbohydrates (g)	276 ± 85.1	242 ± 91.8	0.13
Carbohydrates % of total daily calories	52.9 ± 5.9	40.3 ± 4.2	**<0.001**
Fat (g)	68.8 ± 29.1	106 ± 47.4	**0.002**
Fat % of total daily calories	29 ± 5.3	39.4 ± 4.1	**<0.001**
Cholesterol (mg)	28.7 ± 25.6	453 ± 186	**<0.001**
Saturated fatty acids (g)	15.4 ± 5.7	43.9 ± 20.5	**<0.001**
SFA % of total daily fat	23.1 ± 3.3	41 ± 3.5	**<0.001**
Monounsaturated fatty acids (g)	26.9 ± 14.1	38.4 ± 17.3	**0.025**
MFA % of total daily fat	36.3 ± 5.8	36.1 ± 2.1	**<0.001**
Polyunsaturated fatty acids (g)	19.6 ± 7.9	15 ± 7	0.05
PFA % of total daily fat	28.5 ± 3.7	14.5 ± 2.6	**<0.001**
Fiber g/1000 kcal	20.8 ± 3.2	10.2 ± 2.3	**<0.001**

SFA = Saturated fatty acids, MFA = Monounsaturated fatty acids, PFA = Polyunsaturated fatty acids. Results are presented as average value ± standard deviation, ^+^ only protocols with a plausible caloric intake of more than 1100 kcal daily were considered, * *p*-value from *t*-test for parametric values (protein, MFA)/Mann-Whitney-U-Test for non-parametric values (energy intake, carbohydrates, fat, cholesterol, SFA, PFA, fiber), significant *p*-values are shown in bold.

## Data Availability

Data are available by the corresponding author upon request.

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
