# Peer review of "Plasma Leptin and Adiponectin after a 4-Week Vegan Diet: A Randomized-Controlled Pilot Trial in Healthy Participants"

_ijerph, 2022, doi:10.3390/ijerph191811370_

Round 1

Reviewer 1 Report (Previous Reviewer 1)

the article is now complete and well structured, as well as interesting

Author Response

Thank you! We are very happy about your response.

Reviewer 2 Report (Previous Reviewer 2)

Overall the quality of the paper has improved.

I have a few comments/suggestions.

1) Methods: They stated this:

"Healthy, normal weight subjects (Body Mass Index (BMI) between 18.5 and 30 kg/m2 )"

A BMI of 30 is not considered normal. BMI 25 to 30 is considered overweight. This needed to be explained.

2) Tables: Titles should not have any statistics. Statistics should be given in the footnote of each table. This is very odd.

3) Tables are stand-alone and self-sustaining. That means a reader should understand the data presented in the table without referring to the text of the manuscript. So, please add more footnotes at the bottom of the table with appropriate superscripts embedded in the text of the table. Also, the table footnote should contain the type of statistical tests used, abbreviations used, whether the data were mean ± SD or SE, and the significance level.

4) Data should be plural. So it should be "data are or were" not 'data is or was". Change this throughout the paper.

5) BMI is expanded twice. Once in line 47 and the second time in 87. In fact, BMI is so commonly used acronym, there is no need to expand at the first mention. You can use BMI right away without expanding it. 

6) This should be removed from the statistical analysis section because you mentioned that you have removed the correlation from the paper.

"Correlation analysis was calculated using Spearman-Rho."

7) You stated this:

"Bonferroni correction was performed for all tests."

How did you do this? The correction was done for multiple comparisons? If that is the case, what is the p-value that was used for multiple comparisons?

8) For non-significant p-values, use only 2 decimals.

9) When there is data of 3 or more digits, no need to use the decimal. For example, what is the point of decimal in these: "2240.8 ± 894.8". The value is so large, the decimal is minuscule here. So, convert to 2241 and 895. Do this for all similar expressions.

Author Response

Thank you again for your helpful concerns, which improved the manuscript once again! Manuscript changes are highlighted in yellow.

Overall the quality of the paper has improved. I have a few comments/suggestions.

1) Methods: They stated this:

"Healthy, normal weight subjects (Body Mass Index (BMI) between 18.5 and 30 kg/m2 )"

A BMI of 30 is not considered normal. BMI 25 to 30 is considered overweight. This needed to be explained.

Response: Yes, you are right and the wording is misleading. We reworded to “normal-weight or slight-overweight subjects”.

2) Tables: Titles should not have any statistics. Statistics should be given in the footnote of each table. This is very odd.

Response: We transferred statistical details as well as abbreviation to the footnote. Thank you for this suggestion, which improved the presentation of our tables.

3) Tables are stand-alone and self-sustaining. That means a reader should understand the data presented in the table without referring to the text of the manuscript. So, please add more footnotes at the bottom of the table with appropriate superscripts embedded in the text of the table. Also, the table footnote should contain the type of statistical tests used, abbreviations used, whether the data were mean ± SD or SE, and the significance level.

Response: We moved the additional information (statistic e. g.) to the footnotes and added a short summary of the results of the table to the table legend to support the self-sustaining presentation.  

4) Data should be plural. So it should be "data are or were" not 'data is or was". Change this throughout the paper.

Response: Changed.

5) BMI is expanded twice. Once in line 47 and the second time in 87. In fact, BMI is so commonly used acronym, there is no need to expand at the first mention. You can use BMI right away without expanding it.

Response: We agree with you that BMI is a really familiar acronym. Therefore, we changed the presentation according to your suggestion.

6) This should be removed from the statistical analysis section because you mentioned that you have removed the correlation from the paper.

"Correlation analysis was calculated using Spearman-Rho."

Response: Deleted.

7) You stated this: "Bonferroni correction was performed for all tests."

How did you do this? The correction was done for multiple comparisons? If that is the case, what is the p-value that was used for multiple comparisons?

Response: Thank you for this important concern! We are very sorry for missing the deletion of this sentence. The Bonferroni correction was performed for the target parameters of the main study, but not for the exploratory analyses of adipokine. We deleted the sentence.

8) For non-significant p-values, use only 2 decimals.

Response: We changed the presentation according to your suggestion.

9) When there is data of 3 or more digits, no need to use the decimal. For example, what is the point of decimal in these: "2240.8 ± 894.8". The value is so large, the decimal is minuscule here. So, convert to 2241 and 895. Do this for all similar expressions.

Response: We changed the presentation according to your suggestion.

Reviewer 3 Report (Previous Reviewer 3)

The authors followed the suggestions of the reviewers.

Author Response

Response: Thank you for your time and your helpful concerns!

Round 2

Reviewer 2 Report (Previous Reviewer 2)

I have gone over the paper. By and large paper is ready to go for publication excpet for a few minor revisions.

1) Please change to (line 87) "Criteria of inclusion were: healthy, normal-weight or overweight subjects (BMI between 18.5 and <30 kg/m2 ),....."

There is no such thing as "Slight Overweight" clinically. In fact, they were overweight clinically becuase the BMI was less than 30.

This is a must revision.

2) Please remove 0 from these expressions:

12.10 to 12.1; 1.0 to 1; 130.0 to 130, 0.010 to 0.01, etc. Please do this throuhgout the paper. 0 here has no value whatsoever.

3) Table titles: I made this comment before but they did not follow through.

Table title on the top should be very brief. The data should be clarified in the footnote of the table (sample size, data presentation, mean, SD or SE, etc; statistical tests used, etc). In the footnote, they should give superscripts and those superscripts should be cited in the body of the table.

Authors should reivew some papers in the literature to be familiar with the layout of the table.

This is a must revision

4) Line 284: Please change "Vitamin B12" to "vitamin B-12". A word in the middle of a sentence should not have a "uppercase" although proper nowns require uppercase. However, vitamin should not have uppercase "V'.

So please fix here and elsewhere.

Author Response

Thank you again for your review (I do not get tired of repeating our thank-you as your concerns really improved our manuscript)! The revised parts are again highlighted in yellow.

I have gone over the paper. By and large paper is ready to go for publication excpet for a few minor revisions.

1) Please change to (line 87) "Criteria of inclusion were: healthy, normal-weight or overweight subjects (BMI between 18.5 and <30 kg/m2 ),....."

There is no such thing as "Slight Overweight" clinically. In fact, they were overweight clinically becuase the BMI was less than 30.

This is a must revision.

Response: Changed to "overweight".

2) Please remove 0 from these expressions:

12.10 to 12.1; 1.0 to 1; 130.0 to 130, 0.010 to 0.01, etc. Please do this throuhgout the paper. 0 here has no value whatsoever.

Response: Changed.

3) Table titles: I made this comment before but they did not follow through.

Table title on the top should be very brief. The data should be clarified in the footnote of the table (sample size, data presentation, mean, SD or SE, etc; statistical tests used, etc). In the footnote, they should give superscripts and those superscripts should be cited in the body of the table.

Authors should reivew some papers in the literature to be familiar with the layout of the table.

This is a must revision

Response: I'm really sorry that I misunderstood your suggestion in the last revision. I hope that the current revision is now suitable.

4) Line 284: Please change "Vitamin B12" to "vitamin B-12". A word in the middle of a sentence should not have a "uppercase" although proper nowns require uppercase. However, vitamin should not have uppercase "V'.

So please fix here and elsewhere.

Response: Fixed.

This manuscript is a resubmission of an earlier submission. The following is a list of the peer review reports and author responses from that submission.

Round 1

Reviewer 1 Report

Authors well described results of a pilot RTD concerning changing in plasma leptin and adiponectin after a 4-week vegan diet versus a meat-rich diet. 

I have only 2 suggestions:

  1. change the abbreviation of the meat-rich diet to MrD or otherwise, as MD can easily be confused with the Mediterranean diet.
  2. present the composition of the 2 diets (tab.10) as follows:
    • kcal
    • kcal/kg body weight
    • proteins (g/kg body weight) AND NOT total g daily
    • carbohydrates (g)
    • carbohydrates % of total daily calories
    • fat (g)
    • fat % of total daily calories
      also showing the % of SFA; MUFA and PUFA on total daily fats

I would also recommend adding fibre g/1000kcal as I imagine this is significantly higher in the vegan diet and may contribute to the different results.

It would be interesting to also evaluate the differences in the diets in the two sexes to better discuss the results. 

Reviewer 2 Report

The authors investigated the relationship between a vegan diet and a meat diet in leptin and adiponectin concentrations. The data for this study was extracted from the previous study. There are several issues with this study.

1)  a) First let me state the minor points. The acronyms used are not appropriate such as VD and MD. The VD was popular for vitamin D or venereal disease. The MD is for a physician. So please unabbreviate these 2 phrases, just simply write vegan diet or meat diet. The use of these acronyms is very distractive for readers and they will have a tough time connecting these mundane acronyms with the actual phrase/meaning.

b) Refer 'male to men' and "female to women" throughout the paper.

Now, these are out of my chest, let's come to the most important matter/s.

2) To begin with all these subjects were omnivores at the beginning of the study. Why the meat diet? The authors should have continued with the meat diet as it is. Prior to the meat diet, they were consuming meat anyway and what is the point of them labeling as meat-eaters. 

3) It was surprising that these authors have conducted such a big human study just for 2 adipokines? The energy metabolism and regulation of adiposity are much more complex than these authors make out to be. I am sure they have measured other adipokines such as TNF-∝, ILs, resistin, SAA, etc. If they did not, I would encourage them to measure and give a comprehensive picture of the differences between a vegan diet and a meat diet.

3) The role of diet in leptin and adiposity is a complex one. The role of leptin in hunger and adiposity is very complex. High leptin can be a good thing if a subject does not have leptin resistance as it can reduce hunger, adiposity and increase energy expenditure. So clearly placing leptin on the "bad guys list" is physiologically incorrect. Yes, in leptin-resistant individuals, it might contribute to inflammation. Not knowing how many of the subjects are leptin resistant in this study, it is difficult to say whether diet has any influence on the leptin concentrations.

3) The authors say that they have collected the blood samples from the subjects in a fasting state. But they did not indicate how many hours of fasting.

4) Differences between men and women in leptin concentrations needed to be addressed. In fact, men tend to have less leptin compared to women (Nutrition and Metabolism. 2009 Jan 14;6:3. doi: 10.1186/1743-7075-6-3). I am not sure, clumping all men and women into one category is the right approach given the huge differences in leptin concentrations been the sexes. Subgroup analysis should be a prudent thing to do. From the beginning, they should have been segregated men and women and performed a separate analysis.

5) Correlation is not a good statistical tool to measure the association.

6) Sample size: This is a serious concern because when you have such a small sample size the results are unreliable, difficult to reproduce, and can not be applied to the population at large.

7) I wish they had done the sample size determination.

8) They made a generalized statement on "adipokines" in the conclusion when they only measured 2 markers.

Reviewer 3 Report

Lederer at al. investigate the effects of four-weeks of a vegan diet or meat-rich diet on serum leptin and adiponectin concentrations in female and male. The researchers used statistical analyses to examine the influence of sex, age, and other parameters on the responses of leptin and adiponectin to the dietary interventions. There are demonstrated many correlations between leptin and adiponectin and other variables with a significant statistical power, but the relevance or importance of these correlations is debatable. Sex differences in adipokine response to vegetarian diet have been previously investigated and demonstrated. Leptin and adiponectin concentrations are routinely determined in several laboratories and there are many published studies. The authors could study other less investigated cytokines which increased the value of the study.

The manuscript is overall well written and  statistical methods are complex. I am concerned the authors overanalyze the data resulting from a significant statistical power.

How the authors check the adherence to diet of subjects?

Are you convinced of the adherence to diet of the subjects,  duration of the vegan and meat-rich diet, low number of participants?